# Safe essential scalar-tensor theories

Benjamin Knorr 🆔

Nordita, Stockholm University and KTH Royal Institute of Technology
Hannes Alfvéns väg 12, SE-106 91 Stockholm, Sweden
benjamin.knorr@su.se

October 20, 2023

## Abstract

We discuss the renormalisation group flow of all essential couplings of quantum gravity coupled to a shift-symmetric scalar field at fourth order in the derivative expansion. We derive the global structure of the phase diagram, and identify a bounded region in theory space which is both asymptotically safe in the ultraviolet, and connects to standard effective field theory in the infrared. Our system thus satisfies the weak-gravity bound. The allowed infrared behaviour of the essential four-scalar coupling is restricted by requiring an ultraviolet completion. This bound can be saturated by a theory without free parameters, which gives a concrete example for a fully predictive scalar-tensor theory.

## 1 Introduction

Bringing together quantum theory and dynamical gravity is one of the most difficult open problems in theoretical physics. Despite about a century of effort, no experimentally verified

theory has yet crystalised. At the same time, we now live in a very exciting time where experimental and observational precision comes closer to the one needed to find potential violations of General Relativity, which might originate from quantum gravity effects. This includes the detection of gravitational waves [1], mapping out the event horizon of black holes [2] and high precision cosmological observations [3].

If one thing is however clear on the path of finding an ultimate theory that describes our Universe, it is that a quantisation of pure gravity will not suffice. This relates to the simple observational facts that the Universe contains matter, and that all matter interacts gravitationally. Any pure quantum gravity theory can thus only ever be a small (albeit very important) step towards this goal.

A particularly minimalistic approach towards a theory of quantum gravity and matter is the so-called Asymptotic Safety programme [4, 5]. It stipulates that gravity and Standard Model matter (plus potential additional matter fields to account for *e.g.* dark matter) can be consistently formulated as a non-perturbative quantum field theory by an interacting renormalisation group fixed point. In recent years, the evidence in favour of this scenario has piled up, both in pure gravity scenarios [6–54], and in coupled gravity-matter systems [55–78]. Phenomenological implications of asymptotically safe gravity have been discussed in [77,79–96]. The approach is reviewed *e.g.* in [97–103], and some of the open problems are discussed in [104, 105].

An important observation within this approach (and as a matter of fact for any approach that relies on renormalisation group concepts) is that the couplings of certain pure matter interaction terms cannot vanish at this fixed point [106]. This is because gravitational fluctuations necessarily induce them. Concrete examples for such operators are powers of standard kinetic terms. It is thus of central importance to investigate these couplings, and to check how they influence the existence and the properties of the fixed point. A growing body of works has investigated some of these couplings [60, 64, 73, 75, 76, 92, 106–110], with mixed results: some of them find that the fixed point still persists, others find that the fixed point ceases to exist altogether, or that the allowed number of matter fields is bounded. It is thus central to obtain more data to make a conclusive statement about the fate of the fixed point. An early perturbative study of this system can be found in [111].

One aspect that so far has not been explored in a lot of detail in these investigations is the concept of essential and inessential couplings. Essential couplings appear in observables whereas inessential couplings can be reabsorbed by field redefinitions, or equivalently by imposing certain renormalisation conditions. As a matter of fact, the condition of Asymptotic Safety itself only has to apply to essential couplings [4], so the distinction is crucial – since inessential couplings do not appear in observables, it is irrelevant whether they possess a fixed point. In the context of non-perturbative renormalisation group techniques, this issue has recently received more attention [52, 112]. The practical outcome of this development is a renormalisation group equation purely in terms of essential couplings. Invariably, a discussion of the fixed point in terms of the essential couplings will bring us closer to the final answer whether Asymptotic Safety is realised in gravity.

Another aspect that we want to highlight is the recent technical advancements in the computation of non-perturbative renormalisation group equations. It is now routinely possible to include terms with four [50] and even six [113] derivatives acting on the metric, and to calculate the renormalisation group flow of these couplings without further approximations on a standard laptop.

In this work, we combine the recent developments within the essential scheme and the computational techniques to gain new insights about quantum gravity coupled to a shift-symmetric scalar field. While being a toy model for the real world, it is the simplest possible setup to test how the essential couplings behave in a gravity-matter system. Moreover,

shift-symmetric scalar fields are of interest in string theory in the context of the distance conjecture [114–117].[1] Concretely, we will study this system in a derivative expansion up to quartic order, which includes all local operators up to four derivatives. In section 2 we recall the basics of the non-perturbative renormalisation group equation within the essential scheme that we employ, and specify the concrete approximation that we will use. This is followed by a discussion of the global fixed point structure, with a particular focus on the phenomenologically most interesting regime and a comparison to previous results in section 3. In section 4 we summarise our results, and give a short outlook. Appendix A contains the polynomial from which all relevant fixed point quantities can be computed.

## 2  Setup

In this section we present our setup. We first briefly remind the reader about the functional renormalisation group and its recent incarnation within the essential scheme, and then we display our ansatz for the scalar-tensor theory we are investigating.

### 2.1  Functional renormalisation group

Our investigation is based on the functional renormalisation group (FRG). The FRG revolves around the so-called effective average action $\Gamma_k$, which is a family of actions parameterised by a momentum scale $k$ that smoothly interpolates between a microscopic action for $k \to \infty$, and the standard effective action for $k \to 0$. The argument of $\Gamma_k$ is the expectation value of the bare field, $\Phi = \langle \phi \rangle$. The dependence on $k$ is dictated by the flow equation [118–120]

$$\dot{\Gamma}_k \equiv \partial_t \, \Gamma_k = \frac{1}{2} \mathrm{Tr} \left[ \left( \Gamma_k^{(2)} + \mathfrak{R}_k \right)^{-1} \partial_t \mathfrak{R}_k \right]. \tag{1}$$

In this equation, $t = \ln(k/\Lambda_{\mathrm{UV}})$ is the logarithmic RG scale in units of a reference scale $\Lambda_{\mathrm{UV}}$, $\mathfrak{R}_k$ is a regulator that acts as a momentum-dependent mass term, $\Gamma_k^{(2)}$ is the second functional derivative of $\Gamma_k$ and Tr indicates a functional supertrace, that is a sum over discrete indices and an integration over continuous variables, with an additional minus sign for fermions. Reviews on the FRG can be found in *e.g.* [101, 121–125].

From the flow equation, one can extract the scale dependence of the couplings of a given theory. Concretely, the so-called beta functions are differential equations that govern this dependence. Typically, the beta functions relate to the dimensionless version of the couplings, which is achieved by rescaling the couplings with an appropriate power of the RG scale $k$. Of particular interest are fixed points, which are zeros of the beta functions. They give important information about the phase diagram of a theory, and are intimately linked to second order phase transition. The universality class attached to a given fixed point is then characterised by universal critical exponents $\theta$, which are defined as (minus) the eigenvalues of the matrix of partial derivatives of the beta functions with respect to the couplings, evaluated at the fixed point. A positive critical exponent indicates a relevant coupling, whereas a negative critical exponent is related to an irrelevant coupling. A predictive theory can only have finitely many relevant couplings, since these have to be fixed by experiment to uniquely specify the theory. We will call an asymptotically safe fixed point any fixed point where not all couplings vanish and only a finite number of critical exponents are positive.

---

[1]I would like to thank Ivano Basile for pointing out this connection.

## 2.2 Essential scheme

The flow equation (1) does not discriminate between essential and inessential couplings. In this context, the value of an inessential coupling can be changed by imposing different renormalisation conditions without changing physical observables. Recently, a modification to the flow equation (related to the "standard" renormalisation scheme) has been proposed that distinguishes essential from inessential couplings [52,112], based on earlier work on general flow equations [122, 126] and the technique of bosonisation [127, 128]. The basic idea is that an arbitrary field redefinition of a set of fields $\phi^A$ (the bare fields that appear in the path integral) cannot change observables, if the field redefinition fulfils some criteria. In particular, it should be bijective, respect the symmetries, and not introduce or remove degrees of freedom, but this list is not exhaustive.

An important fact about the FRG is that even if we would remove inessential operators at a given scale $k_1$, upon integrating out modes and lowering the scale to $k_2 < k_1$, we generically reintroduce the inessential operators into the effective average action. To circumvent this, we need a $k$-dependent field redefinition, $\phi^A \mapsto \phi_k^A$. At the level of the FRG, this introduces two extra terms via the inclusion of what has been termed the RG kernel,

$$\Psi^A(\Phi^{\cdot}) = \langle \partial_t \phi_k^A \rangle. \tag{2}$$

Under such a field redefinition, the flow equation changes to

$$\partial_t \Gamma_k + \Psi^A \frac{\delta \Gamma_k}{\delta \Phi^A} = \frac{1}{2} \text{Tr} \left[ \left[ \left( \Gamma_k^{(2)} + \mathfrak{R}_k \right)^{-1} \right]^{AB} \left\{ \partial_t \delta_B{}^C + 2 \frac{\delta \Psi^C}{\delta \Phi^B} \right\} [\mathfrak{R}_k]_{CD} \right]. \tag{3}$$

Here, we used deWitt super-indices to indicate how terms are contracted.

The key insight is that, instead of defining a field redefinition of the bare field and computing the RG kernel via the expectation value (2), we can parameterise $\Psi$ in a bootstrap fashion to remove all inessential couplings for all $k$ directly at the level of $\Gamma_k$. It is however not clear, given any arbitrary $\Psi$, that there is a corresponding field redefinition implementing (2). For this work, we will not deal with this issue, and simply assume that such a field redefinition can indeed be found for our choice of $\Psi$.

The coefficients parameterising $\Psi$ in terms of a given set of operators are called gamma functions. The final set of RG equations is then comprised of the beta functions of the essential couplings, and the gamma functions related to the inessential couplings. Notably, all of them only depend on the essential couplings, and no inessential coupling appears anywhere.

We note in passing that the RG kernel $\Psi$ can mix different fields. We will see in our system that this is in fact necessary if one wants to remove all inessential couplings. This generally makes the insertion $\delta \Psi / \delta \Phi$ in the trace in (3) non-diagonal.

Before we move on to the specific model that we want to investigate, let us discuss some aspects of essential schemes. First, in setting up such a scheme and imposing renormalisation conditions on inessential couplings, one restricts theory space to a subspace. This is clear because one needs a stable notion of what the equations of motion are (in a given approximation scheme), and what (within the same approximation scheme) determines the dynamics. For example, we are interested in a gravity-scalar system without additional propagating degrees of freedom. In this subspace, we can consistently set any momentum-dependent corrections to the propagator to zero by a momentum-dependent field redefinition of the field. If one is instead interested in, *e.g.*, Stelle gravity coupled to a scalar, this is not possible, since in that case the quadratic curvature terms determine the dynamics. That also means that the perturbative fixed point of Stelle gravity lies outside the subspace studied here. For the scalar field, this is even more straightforward: even though the wave function renormalisation is inessential, it cannot be put to zero, but only to an arbitrary positive value. Note that within

a renormalisation group flow, such a restriction to a subspace of theory space is not specific to an essential scheme. Even in a "standard" scheme, we have to make certain assumptions about the pole structure of the propagator in order to define a regulator that actually performs its duty and regulates all infrared divergences. This observation leads us to a bootstrap to distinguish essential from inessential operators in practice: write down the operators that determine the pole structure, and use the resulting equations of motion to identify inessential operators. For our case, this entails that any operator that contains the Ricci scalar or the Ricci tensor is inessential, as is any term containing $D^2\phi$. We then set any inessential term (except the terms that specify the dynamics) to zero.

Second, as a peculiarity of gravity, technically only the combination $G_N\Lambda$ is essential, but nevertheless, also Newton's coupling needs a fixed point in Asymptotic Safety [129]. This means that we have some degree of, but not complete freedom in our choice of essential and inessential couplings in gravity. This freedom is clearly restricted by the fact that we want a positive Newton's constant, but we could have either a positive, negative, or vanishing cosmological constant. Here, we will work with the simplest case and follow previous literature [52] in fixing the running cosmological constant in such a way that $\Lambda = 0$ in the infrared. This avoids off-shell singularities due to not expanding about a solution to the equations of motion. Going consistently beyond this special case requires an $f(R)$-type computation, and we plan to investigate this elsewhere.

## 2.3  Ansatz

Let us now present the ansatz for the effective average action $\Gamma_k$ and the RG kernel $\Psi$ that we will use to solve the flow equation (3) approximately. We will include all terms with up to four derivatives in a derivative expansion. At this order, the essential part of the action[2] reads

$$\Gamma_k = \int \mathrm{d}^4x \,\sqrt{g}\left\{\frac{1}{16\pi G_{N,k}}\left[-2\Lambda_k + R + G_{\mathfrak{E},k}\mathfrak{E}\right] + \frac{1}{2}D^\mu\phi D_\mu\phi + G_{D\phi^4,k}\left(\frac{1}{2}D^\mu\phi D_\mu\phi\right)^2\right\}. \quad (4)$$

Here, $g$ is the metric, $R$ its Ricci scalar, $\mathfrak{E}$ is the four-dimensional Gauss-Bonnet integrand, related to the topological Euler characteristic, and $\phi$ is a shift-symmetric scalar field. We will complete the basis of tensor monomials with the following set of terms:

$$\left\{R^2, S^{\mu\nu}S_{\mu\nu}, \frac{1}{2}(D_\mu D_\nu\phi)^2, R(D_\mu\phi)^2, S^{\mu\nu}D_\mu\phi D_\nu\phi\right\}. \quad (5)$$

In this, $S_{\mu\nu}$ is the tracefree Ricci tensor. Note how all of these operators are proportional to the equations of motion following from (4) (up to shifts in the essential couplings, and higher order terms that we consistently neglect), and thus are indeed related to inessential couplings. Since the action (4) is invariant under diffeomorphisms, we have to add a gauge fixing term. For this, we have to use the background field formalism, splitting the full metric into an arbitrary background and fluctuations around it,

$$g_{\mu\nu} = \bar{g}_{\mu\nu} + h_{\mu\nu}. \quad (6)$$

Other options have been explored in *e.g.* [22, 24, 26, 27, 33, 36, 45, 57, 130–136]. We then follow [52] and use the harmonic gauge,

$$\Gamma_{\mathrm{gf}} = \frac{1}{32\pi G_{N,k}}\int \mathrm{d}^4x \,\sqrt{\bar{g}}\,\bar{g}^{\alpha\beta}\left(\bar{D}^\gamma h_{\alpha\gamma} - \frac{1}{2}\bar{D}_\alpha h\right)\left(\bar{D}^\delta h_{\beta\gamma} - \frac{1}{2}\bar{D}_\beta h\right). \quad (7)$$

---

[2]In the minimal essential scheme related to the standard Gaussian fixed point that we implement [52], essential operators can be identified by using the vacuum equations of motion. If a given operator (excluding the standard kinetic term) vanishes on-shell, it is related to an inessential coupling. For example, in our system the coupling corresponding to $S^{\mu\nu}D_\mu\phi D_\nu\phi$ is inessential since the tracefree Ricci tensor $S$ vanishes in the vacuum. Due to our scale-dependent field redefinition, this is also true along the flow, since we have projected onto the specific subspace of theory space where there are no additional propagating modes.

The gauge fixing gives rise to a standard ghost contribution of the form

$$\Gamma_c = \frac{1}{\sqrt{G_{N,k}}} \int d^4x \sqrt{\bar{g}} \, \bar{c}^\mu \left[ \bar{\Delta} \delta_\mu{}^\nu - \bar{R}_\mu{}^\nu \right] c_\nu \,. \tag{8}$$

Regarding the regularisation, we deviate from [52] and include a suitable endomorphism in all sectors [50]. This choice tremendously simplifies the computation, and minimises the complexity of the RG flow. Concretely, this entails

$$\Delta S_k^h = \frac{1}{64\pi G_{N,k}} \int d^4x \sqrt{\bar{g}} \, h_{\mu\nu} \left[ \mathcal{R}_k^{\mathrm{TL}}(\bar{\Delta}_2) \Pi^{\mathrm{TL}\mu\nu\rho\sigma} - \mathcal{R}_k^{\mathrm{Tr}}(\bar{\Delta}) \Pi^{\mathrm{Tr}\mu\nu\rho\sigma} \right] h_{\rho\sigma} \,,$$
$$\Delta S_k^\phi = \frac{1}{2} \int d^4x \sqrt{\bar{g}} \, \phi \, \mathcal{R}_k^\phi(\bar{\Delta}) \phi \,, \tag{9}$$
$$\Delta S_k^c = \frac{1}{\sqrt{G_{N,k}}} \int d^4x \sqrt{\bar{g}} \, \bar{c}^\mu \, \mathcal{R}_k^c(\bar{\Delta}_c) c_\mu \,.$$

The second variation of $\Delta S_k$ then defines the regulator $\mathfrak{R}_k$ in (3). Here, we used the operators

$$\bar{\Delta} = -\bar{D}^2 \,,$$
$$\bar{\Delta}_2^{\mu\nu\rho\sigma} = \left( \bar{\Delta} - \frac{2}{3}\bar{R} \right) \Pi^{\mathrm{TL}\mu\nu\rho\sigma} - \bar{C}^{\mu\rho\nu\sigma} - \bar{C}^{\nu\rho\mu\sigma} \,, \tag{10}$$
$$\bar{\Delta}_c^{\mu\nu} = \bar{\Delta} \, \bar{g}^{\mu\nu} - \bar{R}^{\mu\nu} \,,$$

where $\bar{C}$ is the background Weyl tensor, and $\Pi^{\mathrm{TL}}$ and $\Pi^{\mathrm{Tr}}$ indicate the traceless and trace projectors built from the background metric, respectively. The endomorphisms are chosen in such a way that if we would set the regulator shape functions $\mathcal{R}_k$ to be minus their argument, only the cosmological constant term would remain in $\Gamma_k^{(2)} + \mathfrak{R}_k$. This significantly simplifies the computation, and reduces the number of propagators appearing in all beta and gamma functions. For the concrete results presented in the next section, we will use the linear cutoff [137, 138],

$$\mathcal{R}_k(x) = (k^2 - x)\theta(1 - x/k^2) \,, \tag{11}$$

where $\theta$ is the Heaviside distribution.

Next, we present the RG kernels that we employ. They read

$$\Psi_{\mu\nu}^g = \gamma_g \, g_{\mu\nu} + \gamma_R R \, g_{\mu\nu} + \gamma_S \, S_{\mu\nu}$$
$$+ \left( \gamma_{gD\phi^2} - \frac{1}{4}\gamma_{D\phi D\phi} \right) g_{\mu\nu} D^\alpha \phi D_\alpha \phi + \gamma_{D\phi D\phi} D_\mu \phi D_\nu \phi \,, \tag{12}$$
$$\Psi^\phi = \gamma_\phi \, \phi + \gamma_{\Delta\phi} \Delta\phi \,.$$

The gamma functions depend on $k$, but we suppress this dependence to avoid cluttered notation. We have split all terms into trace and traceless part for convenience. These seven gamma functions can be used to fix the inessential couplings related to (5), the cosmological constant and the scalar wave function renormalisation. Note that we cannot add *e.g.* a term $\propto R\phi$ to $\Psi^\phi$ even though it would be of the right derivative order, since this would break the shift symmetry of our action.

Let us now introduce the dimensionless couplings that will be used to analyse the phase diagram. For the couplings in (4), we define

$$g = G_{N,k} k^2 \,, \qquad \lambda = \Lambda_k k^{-2} \,, \qquad g_{\mathfrak{E}} = G_{\mathfrak{E},k} k^2 \,, \qquad g_{D\phi^4} = G_{D\phi^4,k} k^4 \,. \tag{13}$$

Here, we suppress the $k$-dependence of the dimensionless couplings. Sometimes, instead of $g_{\mathfrak{E}}$, we will use the dimensionless coupling $\Theta$ defined as

$$\Theta = \frac{G_{\mathfrak{E},k}}{16\pi G_{N,k}} \,. \tag{14}$$

Some of the gamma functions also carry a mass dimension. We hence introduce

$$\hat{\gamma}_R = \gamma_R k^2 \,, \quad \hat{\gamma}_S = \gamma_S k^2 \,, \quad \hat{\gamma}_{gD\phi^2} = \gamma_{gD\phi^2} k^4 \,, \quad \hat{\gamma}_{D\phi D\phi} = \gamma_{D\phi D\phi} k^4 \,, \quad \hat{\gamma}_{\Delta\phi} = \gamma_{\Delta\phi} k^2 \,. \tag{15}$$

Finally, to implement the *minimal* essential scheme [52], with the linear regulator (11) we have to set

$$\lambda = \frac{3}{16\pi} g \,. \tag{16}$$

This corresponds to a vanishing physical cosmological constant $\Lambda \equiv \Lambda_0 = 0$ in the infrared (IR). The different factor compared to [52] originates from the additional contribution from the scalar field.

## 3 Structure of the RG flow

In this section we discuss our results. We first present the complete fixed point structure, and then zoom in on the phenomenologically most interesting region. Finally, we compare our results to the literature. The beta and gamma functions have been computed with the help of the Mathematica package *xAct* [139–144], and the code is based on the recent technical developments presented in [50, 113]. The completely evaluated right-hand side of (3) for general regulator shape functions is provided in a Mathematica notebook [145], together with some of the results for the shape function (11).

### 3.1 Fixed point structure

Our system features two essential couplings, $g$ and $g_{D\phi^4}$,[3] and their beta functions are rational functions of these two couplings. As a consequence, we are able to determine *all* fixed points of this system by relating them to the roots of a specific polynomial obtained by computing a Gröbner basis [146]. Such a high degree of analytical control over the phase diagram is unusual when higher order operators are taken into account, but it arises as a key simplification due to the minimal essential scheme.

We find a total of 45 fixed points. First, there is the Gaussian fixed point,

$$g = g_{D\phi^4} = 0 \,, \qquad \theta_1 = -4 \,, \qquad \theta_2 = -2 \,. \tag{17}$$

At this fixed point, all gamma functions vanish by construction. Next, there are two pure matter fixed points,

$$
\begin{aligned}
g &= 0 \,, & g_{D\phi^4} &= -\frac{64\pi^2}{5}\left(45 \pm \sqrt{1945}\right) \,, \\
\theta_1 &= \frac{1}{18}\left(389 \pm 7\sqrt{1945}\right) \,, & \theta_2 &= -\frac{1}{45}\left(125 \pm \sqrt{1945}\right) \,.
\end{aligned}
\tag{18}
$$

The corresponding non-vanishing values of the gamma functions read

$$\gamma_g = \frac{1}{45}\left(35 \pm \sqrt{1945}\right) \,, \qquad \gamma_\phi = -\frac{1}{18}\left(35 \pm \sqrt{1945}\right) \,. \tag{19}$$

---

[3]The coupling of the Gauss-Bonnet term is also essential, but since it does not drive the RG flow, we ignore it in most of the discussion.

| # | $g$ | $g_{D\phi^4}$ | $\theta_1$ | $\theta_2$ |
|---|-----|---------------|-----------|-----------|
| 4 | -1880.46 | -45428.8 | -995.651 | -27.5839 |
| 5 | -217.334 | -452.114 | -25.4916 | 137.818 |
| 6 | -55.8522 | 2317.35 | -194.713 | 42.3717 |
| 7 | -21.4507 | 30588.4 | 83.5776 | 2142.31 |
| 8 | -12.5072 | -96378.1 | -18.8876 | 100.316 |
| 9 | -12.4155 | 54157.2 | -16.8812 | 743.789 |
| 10 | -11.2387 | 2172.73 | -39.3934 | -15.7947 |
| 11 | -9.60693 | -166.377 | -13.0876 | 69.1431 |
| 12=A | 0.251973 | -17.9179 | 0.403505 | 1.99177 |
| 13=B | 0.254186 | -6.03273 | -0.408876 | 1.96731 |
| 14 | 0.384777 | $7.53694 \times 10^6$ | -12138.5 | 3.11094 |
| 15 | 0.435532 | -13572.9 | 3.95756 | 44.8923 |
| 16 | 1.01185 | -59.6739 | -62.4599 | 28.3001 |
| 17 | 1.02905 | 658.299 | -76.2433 | -19.8638 |
| 18 | 1.03507 | 48304.1 | -367.626-98.792 **i** | -367.626+98.792 **i** |
| 19 | 1.08258 | -104976 | -85.2787 | 115.436 |
| 20 | 1.13104 | 8134.46 | -409.849 | -24.5517 |
| 21 | 1.14822 | 4919.19 | -440.194 | 21.1620 |
| 22 | 1.25351 | -73.0455 | -412.705 | -159.974 |
| 23 | 5.85019 | -371.579 | -103.639 | 1361.54 |
| 24 | 6.55425 | -918.002 | 62.8289 | 963.477 |
| 25 | 7.68539 | $-6.25216 \times 10^6$ | 275.704 | 6735.06 |
| 26 | 9.10630 | -90789.3 | -8128.55 | -164.645 |
| 27 | 16.7936 | -240.397 | -5.19338 | 240.628 |
| 28 | 18.1781 | 621.804 | -411.878 | -15.9696 |
| 29 | 20.4463 | -38230.0 | -209.079 | 39.5238 |
| 30 | 20.9106 | -388392 | 47.3196 | 340.879 |
| 31 | 43.5431 | -270.960 | -7.69868 | 260.151 |
| 32 | 200.976 | -336.141 | 12.9708 | 984.115 |
| 33 | 250.856 | -30034.3 | -12.1640 | 413.042 |
| 34 | 318.198 | 2739.26 | -402.063 | -3.54196 |
| 35 | 393.082 | $7.86168 \times 10^6$ | -9204.25 | -6.87014 |
| 36/37 | -111.139±767.092 **i** | 1200.27±3594.62 **i** | 5.89505±7.02388 **i** | 140.718∓534.752 **i** |
| 38/39 | 6.23901±0.84550 **i** | 98.2402∓58.9834 **i** | 365.640∓565.508 **i** | -27.1113∓88.1939 **i** |
| 40/41 | 7.47250±2.56062 **i** | -17836.2±17193.6 **i** | -477.388±1076.058 **i** | 64.249∓218.822 **i** |
| 42/43 | 12.79020±4.96100 **i** | -258.028∓82.480 **i** | 210.616±96.830 **i** | 10.04037∓2.03541 **i** |
| 44/45 | 13.48948±2.63750 **i** | 329.181±181.642 **i** | -194.625∓80.580 **i** | 7.11574±9.59831 **i** |

Table 1: List of fixed point values and critical exponents of all fully interacting fixed points. We omit fixed points 1-3 which are the Gaussian and the two pure matter fixed points, (17) and (18). Fixed points 12 (A) and 13 (B) are discussed in more detail in subsection 3.2.

| # | $\gamma_g$ | $\hat{\gamma}_R$ | $\hat{\gamma}_S$ | $\hat{\gamma}_{gD\phi^2}$ | $\hat{\gamma}_{D\phi D\phi}$ | $\gamma_\phi$ | $\hat{\gamma}_{\Delta\phi}$ |
|---|---|---|---|---|---|---|---|
| 4 | -6.54283 | -4.98652 | 274.224 | 23604.1 | $-6.71945\times10^6$ | 27.0852 | 47.5005 |
| 5 | -0.849483 | -3.02027 | 15.5461 | -4926.80 | -65396.1 | 30.2977 | 3.54137 |
| 6 | 0.505966 | -2.68478 | -10.4378 | -25325.9 | -103858 | 39.7324 | 14.5518 |
| 7 | -2.87382 | -0.920909 | -3.03057 | -89719.0 | -376582 | 76.2986 | 114.096 |
| 8 | -12.8480 | 1.46784 | 31.0350 | -4302.40 | -24508.7 | 2.36784 | 11.1601 |
| 9 | -11.6924 | 1.22153 | 27.2109 | -11907.1 | -53213.5 | 10.8651 | 23.1200 |
| 10 | -10.3844 | 0.812552 | 22.7894 | -2692.78 | -15422.8 | 6.75536 | 7.37795 |
| 11 | -8.30948 | 0.276278 | 15.8305 | -923.553 | -6467.02 | 9.87186 | 3.35805 |
| 12=A | -1.40129 | 0.0453143 | 0.00400744 | 0.394280 | -0.204548 | 0.830224 | 0.00736639 |
| 13=B | -1.39434 | 0.0451921 | 0.00466159 | 0.586224 | 0.000516815 | 0.781908 | 0.00740451 |
| 14 | -0.926237 | -0.0143782 | 0.0858009 | 27175.9 | 27056.3 | 5.14988 | 15.2277 |
| 15 | -0.623084 | -0.0563173 | 0.126834 | 162.995 | 153.668 | -4.43884 | 0.0949211 |
| 16 | -3.92961 | 6.23607 | -0.947775 | 772.570 | -0.717093 | 7.28808 | 2.39666 |
| 17 | -3.57492 | 6.17251 | -0.829853 | 1213.44 | 92.0733 | 5.21384 | 3.27612 |
| 18 | -3.53706 | 5.49323 | -0.687204 | 26710.0 | 6548.59 | 30.7745 | 52.9851 |
| 19 | -2.66547 | 6.15060 | -0.499007 | 4498.37 | 597.828 | 1.87941 | 10.3003 |
| 20 | -1.78969 | 6.95670 | -0.257039 | -9904.79 | -1475.66 | -21.8282 | -20.9845 |
| 21 | -1.52167 | 7.13342 | -0.149243 | -9490.64 | -1207.33 | -24.0310 | -20.5177 |
| 22 | 0.266818 | 8.90468 | 0.660678 | -5843.04 | -92.7204 | -38.4790 | -13.8451 |
| 23 | -3.30141 | -1.90586 | -4.98943 | 1062.30 | 5354.15 | 16.8192 | -14.5002 |
| 24 | -3.15509 | -1.92309 | -5.54848 | 1917.13 | 11174.8 | -1.99631 | -49.8234 |
| 25 | -2.91833 | -1.71407 | -5.59840 | 4428.73 | 10435.7 | 4.13445 | 13.6726 |
| 26 | -2.71089 | -2.07001 | -7.64147 | -1606.21 | 9171.85 | 189.225 | 307.502 |
| 27 | -2.28136 | 0.315077 | 0.526542 | -5251.03 | -32973.5 | 22.0634 | -39.6772 |
| 28 | -3.49927 | 6.87243 | 32.9168 | -14498.7 | -96599.4 | 11.2396 | -103.521 |
| 29 | 0.998401 | -14.2753 | -75.0272 | 2523.86 | 31991.5 | 15.8206 | 32.1711 |
| 30 | 1.38824 | -15.8667 | -83.8412 | -1634.87 | 10163.8 | 4.19319 | 13.7723 |
| 31 | -0.603523 | -2.25571 | -18.7661 | -6196.16 | -39421.7 | 27.7577 | -11.5647 |
| 32 | 9.83412 | 5.58425 | 8.17547 | -17751.3 | -134275 | 81.1446 | -7.78173 |
| 33 | -7.72113 | -8.83462 | -100.413 | -3646.71 | -273766 | -13.5954 | -14.4077 |
| 34 | -1.94679 | -4.34857 | -80.2796 | -19772.5 | -400250 | 15.2520 | -15.8483 |
| 35 | -7.38218 | -7.28930 | -113.435 | 241884 | 668607 | 3.10214 | 12.1512 |
| 36/37 | -1.10468 $\pm2.61209\,\mathbf{i}$ | -3.16827 $\pm0.94324\,\mathbf{i}$ | -2.2828 $\mp118.9778\,\mathbf{i}$ | -5194.4 $\mp27187.4\,\mathbf{i}$ | $1.189336\times10^6$ $\pm35161\,\mathbf{i}$ | 26.9380 $\pm1.5062\,\mathbf{i}$ | -1.8429 $\mp20.3882\,\mathbf{i}$ |
| 38/39 | -3.19699 $\pm0.17219\,\mathbf{i}$ | -1.85166 $\mp0.01021\,\mathbf{i}$ | -5.14750 $\mp0.61867\,\mathbf{i}$ | 1061.019 $\pm899.307\,\mathbf{i}$ | 5439.15 $\pm6022.74\,\mathbf{i}$ | 19.7032 $\mp16.1710\,\mathbf{i}$ | -6.1749 $\mp28.4864\,\mathbf{i}$ |
| 40/41 | -2.96125 $\pm0.33761\,\mathbf{i}$ | -1.47721 $\mp0.54704\,\mathbf{i}$ | -4.47977 $\mp3.10253\,\mathbf{i}$ | -1703.43 $\pm5524.92\,\mathbf{i}$ | -5937.3 $\pm43547.6\,\mathbf{i}$ | 97.9553 $\mp26.1632\,\mathbf{i}$ | 166.439 $\mp39.069\,\mathbf{i}$ |
| 42/43 | -2.17059 $\pm0.32663\,\mathbf{i}$ | -1.28154 $\pm0.63844\,\mathbf{i}$ | -6.65579 $\pm1.37979\,\mathbf{i}$ | -4356.63 $\mp1651.57\,\mathbf{i}$ | -26637.3 $\pm10927.8\,\mathbf{i}$ | 16.9421 $\mp6.8571\,\mathbf{i}$ | -40.3723 $\mp17.8384\,\mathbf{i}$ |
| 44/45 | -2.23153 $\pm0.07375\,\mathbf{i}$ | -0.905157 $\pm0.916267\,\mathbf{i}$ | -4.93951 $\pm3.45345\,\mathbf{i}$ | -5624.62 $\mp2771.01\,\mathbf{i}$ | -34753.7 $\pm18532.7\,\mathbf{i}$ | 6.94760 $\pm0.70019\,\mathbf{i}$ | -69.9096 $\pm3.2036\,\mathbf{i}$ |

Table 2: List of fixed point values of the gamma functions $\gamma_g, \hat{\gamma}_R, \hat{\gamma}_S, \hat{\gamma}_{gD\phi^2}, \hat{\gamma}_{D\phi D\phi}, \gamma_\phi$ and $\hat{\gamma}_{\Delta\phi}$ of all fully interacting fixed points.

Naively, one would only expect one non-trivial pure matter fixed point, since at least perturbatively the beta function for the scalar coupling is quadratic in itself. The occurrence of an additional fixed point is due to higher loop terms included in $\gamma_\phi$, effectively corresponding to the scalar anomalous dimension.

Finally, there are 42 fully interacting fixed points, of which 32 are real, and 10 are complex conjugate pairs. They are related to the roots of a polynomial of order 42 that we have written out in Appendix A. A complete list of these fixed points including their critical exponents is given in Table 1, and the corresponding fixed point values of the gamma functions are given in Table 2. Notably, there is only a single real fixed point with complex conjugate critical exponents. This is to be contrasted with quantum gravity computations in the standard scheme, where complex conjugate critical exponents appear routinely.

Not all of these fixed points are related to well-defined IR and ultraviolet (UV) behaviours. As a matter of fact, only three of the interacting fixed points, together with the Gaussian fixed point, define the region of all trajectories related to globally well-defined fundamental quantum field theories. We will focus on this part of theory space in the next subsection.

## 3.2 The asymptotically safe shark fin

We will now focus on the region of theory space that features both a UV completion by an interacting fixed point, and IR physics governed by the Gaussian fixed point, and which is depicted in Figure 1. This region is bounded by four fixed points and their separatrices: a fully attractive interacting UV fixed point A (purple dot, # 12 in Table 1), a fully interacting fixed point B that is a saddle point (green dot, # 13 in Table 1), an interacting matter fixed point C (orange dot, lower sign in (18)) that is also a saddle point, and the Gaussian IR fixed point D (red dot). The separatrices between these fixed points are shown as black lines, and come in the form of a shark fin. Some trajectories with a completely well-defined RG flow are indicated as dashed grey lines. All trajectories that are connected to these fixed points but are located outside of the asymptotically safe shark fin are either UV or IR divergent, and thus belong to the "Asymptotic Safety swampland" [147]. The dark yellow line indicates where the scalar self-coupling does not flow, and its peak is related to the so-called weak-gravity bound[4] [60, 64, 73, 75, 76, 92, 106, 108–110], which we discuss in subsection 3.3 where we compare our results to the literature.

The existence of fixed point B and the corresponding separatrix BD implies that there is a bound on the IR behaviour of the scalar self-interaction. More concretely, we find that in the IR,

$$k \to 0: \qquad g \sim G_N k^2, \qquad g_{D\phi^4} \sim \left(a + b \ln\left[G_N k^2\right]\right) G_N^2 k^4. \qquad (20)$$

Here, $G_N \equiv G_{N,0}$ is the physical Newton's constant at $k = 0$. The additional logarithmic running of the scalar coupling is solely due to gravitational fluctuations – it disappears on the pure matter separatrix CD. As a matter of fact, its coefficient $b$ is universal in the sense that it does not depend on the shape of the regulator functions. However, it likely depends on the gauge choice, similar to other one-loop logarithms [149, 150]. In our gauge,

$$b = \frac{203}{5}. \qquad (21)$$

---

[4]In its original formulation, the weak-gravity bound is a statement about the existence of the shifted Gaussian fixed point. Given a specific matter system and its fixed point structure, it asks the question what happens to these fixed points when switching on the gravitational couplings (*e.g.* Newton's constant) by hand, without computing the flow of the latter. The weak-gravity bound is then the value of $g$ (and other gravitational couplings) where the Gaussian fixed point (shifted by gravity) collides with another (interacting) fixed point, and indicates that above this critical strength, Asymptotic Safety cannot be realised at the shifted Gaussian fixed point, but has to be more non-perturbative (see however [148] for a recent reinvestigation and refined notion of the weak-gravity bound).

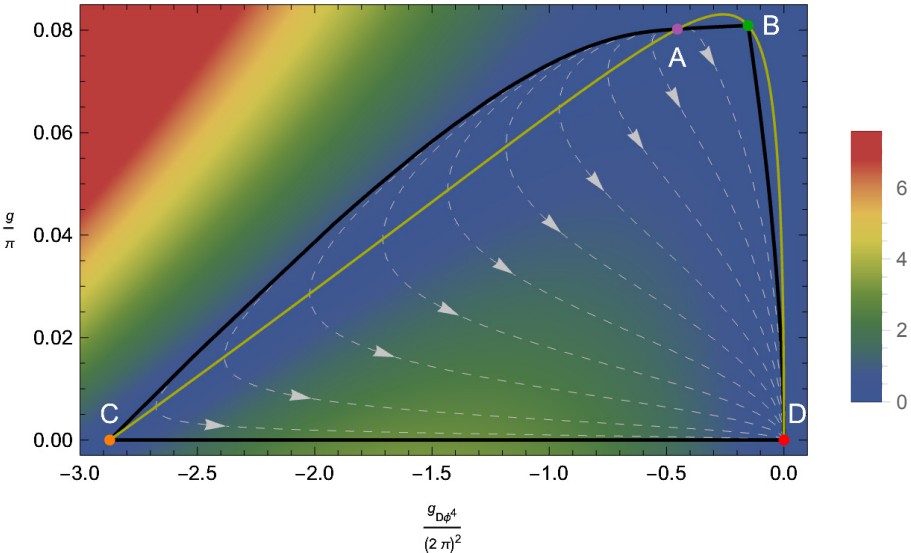

Figure 1: The phase diagram with the four fixed points enclosing all well-defined theories in our system. The four fixed points are the fully interacting UV fixed point A (purple dot) with two relevant directions, the fully interacting fixed point B (green dot) with one relevant and one irrelevant direction, a pure matter fixed point C (orange dot) with one relevant and one irrelevant direction, and the Gaussian IR fixed point (red dot). The black lines indicate the separatrices between these fixed points, whereas selected asymptotically safe trajectories are indicated as dashed grey lines, with arrows pointing towards the IR. The colour scheme indicates the velocity of the RG flow, and is defined as the norm of the vector of beta functions for the rescaled couplings $g/\pi$ and $g_{D\phi^4}/(2\pi)^2$. The dark yellow line indicates where the scalar coupling does not flow, and its peak corresponds to the weak-gravity bound. By construction the RG flow on this line is exactly vertical (or vanishes identically at the fixed points).

The value of $a$ depends on the specific trajectory, and its value for all trajectories[5] is bounded by its value along BD. Concretely, for any trajectory connecting A with D, we find the upper bound

$$a \lesssim 31.85\,. \tag{22}$$

This represents the free parameter in the universality class corresponding to the ultraviolet completion via fixed point A. Such an upper bound for the Wilson coefficient $a$ from the RG flow is also interesting from the point of view of positivity bounds [151]. Combining (22) with the corresponding positivity bound might further reduce the landscape of viable theories.

In Figure 2 we illustrate the running of the two essential couplings along the separatrix BD.[6] One can clearly see the two scaling regimes dominated by the two fixed points, and a transition between the two regimes at around the Planck scale. We note that the theory corresponding to BD has no free parameters: the scalar self-coupling is fixed in terms of Newton's coupling at all scales, and Newton's coupling simply sets the scale of the theory. This is in agreement with the fact that fixed point B only has one positive critical exponent, which is dealt with by the scale setting.

---

[5]This excludes the separatrix CD, where $G_N$ vanishes and $a$ goes to (minus) infinity in such a way that the product $a\,G_N^2$ stays finite. Specifically, along CD we have $g_{D\phi^4} \sim G_{D\phi^4}k^4$ as $k \to 0$. In this way the dimensionful scalar coupling $G_{D\phi^4}$ simply sets the scale instead of $G_N$.

[6]Generic AD trajectories behave qualitatively similar to the separatrix BD. Trajectories connecting A with D that closely pass either B or C have an additional intermediate scaling regime.

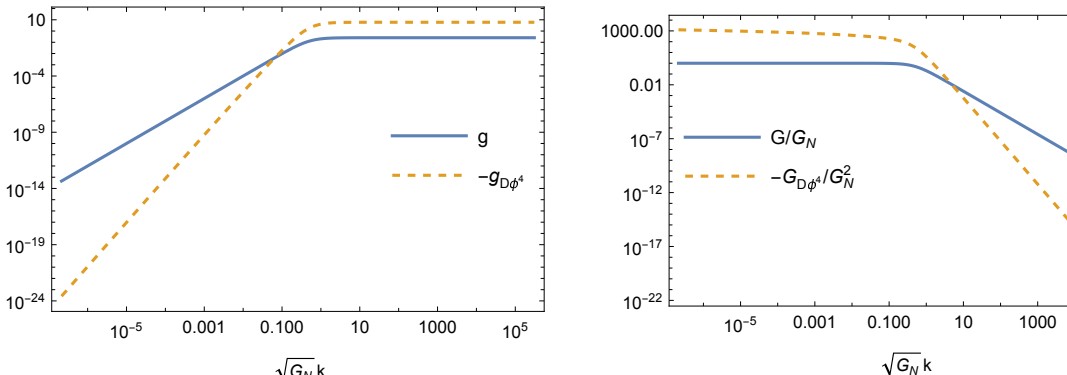

Figure 2: The scale dependence of the essential couplings along the separatrix BD. The left panel shows the dimensionless couplings, whereas the right panel shows the dimensionful couplings. Everything is expressed in units of the Newton's constant $G_N$ defined by (20). The additional logarithmic running of the scalar self-interaction can be clearly seen in the right panel.

Coming to the coupling of the Euler term, we find that

$$
\begin{aligned}
\text{A}: && \dot{g}_{\mathfrak{E}} &= 0.0677789\,, \\
\text{B}: && \dot{g}_{\mathfrak{E}} &= 0.0689797\,.
\end{aligned}
\tag{23}
$$

This shows that at A and B we have the scenario described in [50], namely that the coupling of the topological Euler term goes to infinity in the UV. Correspondingly the path integral would naively be dominated by topologies with a negative Euler characteristic. On the other hand, at C and D, $g$ vanishes, and focussing on the flow of $g_{\mathfrak{E}}$ is somewhat misleading since by our convention it appears together with Newton's constant in the action. Instead, we should focus on the flow of the coupling $\Theta$ which is dimensionless. For this coupling, at $g = 0$ we find the beta function

$$
g \to 0: \qquad \dot{\Theta} = \frac{587}{4800\pi^2} - \frac{1856}{75} \frac{1}{5g_{D\phi^4} + 512\pi^2}\,.
\tag{24}
$$

At the fixed points, we thus find

$$
\begin{aligned}
\text{C}: && \dot{\Theta} &= \frac{4210 - 29\sqrt{1945}}{43200\pi^2}\,, \\
\text{D}: && \dot{\Theta} &= \frac{71}{960\pi^2} = \frac{1}{(4\pi)^2}\left[\frac{53}{45} + \frac{1}{180}\right]\,.
\end{aligned}
\tag{25}
$$

At the Gaussian fixed point D, the beta function (and the corresponding logarithmic running) is universal in the same sense as outlined above for the coefficient $b$ in (21), and we have split the result into the gravitational (first term) and the scalar (second term) contribution.

Let us finally discuss the IR behaviour of the gamma functions. As noted earlier, some of the gamma functions are dimensionless, whereas others are dimensionful. It is clear that the dimensionless gamma functions must go to zero in the IR: by construction, in the minimal essential scheme all gamma functions vanish when all couplings are set to zero, and we do not have any dimensionless couplings that could give them a finite value. On the other hand, the dimensionful gamma functions can be finite. Indeed, we find

$$
\begin{aligned}
k \to 0: \qquad \gamma_R &= -\frac{49}{240\pi}G_N\,, & \gamma_S &= \frac{43}{60\pi}G_N\,, \\
\gamma_{gD\phi^2} &= \frac{37}{10}G_N^2\,, & \gamma_{D\phi D\phi} &= \frac{86}{15}G_N^2\,.
\end{aligned}
\tag{26}
$$

Notably, $\gamma_{\Delta\phi}$ vanishes at the Gaussian fixed point even though it could have had a finite value. The IR behaviour of the remaining gamma functions reads

$$k \to 0: \qquad \gamma_g \sim -\frac{419}{24\pi}G_N k^2\,, \qquad \gamma_\phi \sim \frac{451}{48\pi}G_N k^2\,, \qquad \gamma_{\Delta\phi} \sim \frac{277}{360\pi^2}G_N^2 k^2\,. \qquad (27)$$

These values are determined purely in terms of Newton's constant, and the scalar coupling does not enter. In particular, in the pure matter system, the only two non-vanishing gamma functions $\gamma_g$ and $\gamma_\phi$ have the simple expression

$$g \to 0: \qquad \gamma_g = \frac{8g_{D\phi^4}}{512\pi^2 + 5g_{D\phi^4}}\,, \qquad \gamma_\phi = -\frac{20g_{D\phi^4}}{512\pi^2 + 5g_{D\phi^4}}\,. \qquad (28)$$

All other gamma functions vanish exactly for $g = 0$. This is not entirely surprising: except for $\gamma_{\Delta\phi}$, all gamma functions are related to inessential couplings corresponding to monomials with at least one curvature. On the other hand, $\gamma_{\Delta\phi}$ vanishes in the pure matter theory since our scalar coupling does not induce a non-trivial momentum dependence for the scalar propagator. The only diagram it gives rise to is a tadpole diagram with quadratic momentum dependence, and thus a finite $\gamma_\phi$.

## 3.3   Comparison to previous results

We now briefly compare our results with previous work on quantum gravity coupled to a shift-symmetric scalar field. One should bear in mind that all previous work on the topic has employed the "standard" RG scheme, where essential and non-essential couplings have not been distinguished.

The first discussion of higher order matter terms induced by gravity in the present context is [106], where a setup with all four-scalar interactions was discussed, including terms that are not shift-symmetric. Compared to our work, the quadratic curvature terms as well as the non-minimal curvature-scalar interactions were neglected. Within this setup and next to the Gaussian fixed point D, only the equivalent of the pure matter fixed point C was found. Treating $g$ as a parameter, a (partial) fixed point for the scalar coupling exists for small enough $g$, which ultimately collides with yet another (partial) fixed point at a critical value $g_{\text{crit}}$. This has later been dubbed the weak-gravity bound, and in our system this partial fixed point line is indicated by the dark yellow line in Figure 1. The critical value corresponds to the peak of this curve. In this sense, the setup in [106] does not fulfil the weak-gravity bound as the would-be fixed point value for $g$ is too large. A violation of the weak-gravity bound has also been found in [110] within a similar truncation, but including an arbitrary number of scalar fields. By contrast, our setup fulfils the weak-gravity bound

$$g_{\text{crit}} \approx 0.261\,, \qquad (29)$$

and we find the fixed points A and B of the full system below this critical value for $g$.

The impact of the non-minimal curvature-scalar coupling $R^{\mu\nu}D_\mu\phi D_\nu\phi$ has been investigated in [60], while neglecting all other non-trivial matter couplings. This study did find a suitable fixed point, and no weak-gravity bound has been observed.

Finally, [76] discussed the most elaborate setup, which compared to ours only neglected the quadratic curvature terms and the two-scalar-four-derivative term. Their findings are quite similar to ours, and in particular the weak-gravity bound was respected. The results for the pure matter fixed point are compatible with our fixed point C accounting for the differences in the approximation and regularisation schemes. The completely interacting fixed points show similar features as our fixed points A and B, most importantly the fact that one fixed point has one relevant direction fewer than the other.

To check the effect of truncations, we tried to mimic [106,110] by setting the corresponding gamma functions to zero by hand, but our result remains qualitatively unchanged. We finally note that [60,106,110] employed a different gauge condition, namely the Landau gauge $\alpha \to 0$ with $\beta = 0$, whereas [76] used the harmonic gauge $\alpha = \beta = 1$ that we also used here.

We offer the following tentative conclusion. Due to the generally smaller fixed point values for $g$ in the minimal essential scheme compared to the standard scheme, we can achieve compatibility with the weak-gravity bound more easily, and higher order operators seem to play a less important role. By contrast, in the standard scheme the inclusion of some non-minimal couplings seems to be essential to find a fixed point. We also note that the fixed points A and B are rather close to the weak-gravity bound, indicating that even more elaborate truncations than the present one could result in a violation of the bound. In any case, a deeper understanding of the gauge dependence, the importance of non-minimal couplings and the differences between essential and standard scheme seems necessary to give a final answer regarding the true fixed point structure in this system.

## 4 Summary

We have investigated the non-perturbative renormalisation group flow of quantum gravity coupled to a shift-symmetric scalar field in a systematic derivative expansion. For the first time, the full four-derivative approximation has been completely resolved. Employing the minimal essential scheme, we found evidence for two fully interacting fixed points, both of which provide an ultraviolet completion for the theory, and are connected to infrared physics dictated by the Gaussian fixed point. One of the two fixed points is a saddle, and the renormalisation group trajectory connecting it to the Gaussian fixed point corresponds to a theory without any free parameters. We also derived a universal logarithmic running of the scalar coupling near the Gaussian fixed point, and a bound on the respective Wilson coefficient. It will be interesting to check the compatibility of the latter with the corresponding positivity bound [151].

We briefly discussed the weak-gravity bound, which is respected in our approximation. We however also find that the interacting fixed points are close to the weak-gravity bound, indicating that the inclusion of higher-order terms can still change the overall picture qualitatively. Our results are compatible with parts of the literature that include non-minimal interactions [60,76], but the ultimate fate of the bound is yet unsettled [106,110], see however [148] for a new perspective on the weak-gravity bound.

It will be interesting to study the next order in the derivative expansion in this system. Already in the pure gravitational sector, the inclusion of the well-known Goroff-Sagnotti term [29,152,153] will yield a lot of new insights, but also in the matter sector additional essential couplings related to six-derivative operators occur. Ultimately, one is interested in finding signs for apparent convergence [34] to settle the question whether scalar-tensor theories can be ultraviolet-completed. Similarly, we can straightforwardly apply our setup to other matter fields like the photon, which we anticipate to give us a better understanding of the weak-gravity bounds also in the other matter sectors.

## Acknowledgements

I would like to thank Alessio Baldazzi, Gustavo de Brito, Kevin Falls, Yannick Kluth, Alessia Platania and Marc Schiffer for interesting discussions and collaboration on related projects, and Alessia Platania for useful comments on the manuscript. This work was supported by Perimeter Institute for Theoretical Physics and Nordita. Research at Perimeter Institute is sup-

ported in part by the Government of Canada through the Department of Innovation, Science and Economic Development and by the Province of Ontario through the Ministry of Colleges and Universities. Nordita is supported in part by NordForsk.

## A  Polynomial specifying the fixed point structure

In this appendix we present the polynomial whose roots determine the fully interacting fixed points displayed in Table 1. The fixed point value of $g = \pi q$ is a root of

$$
\begin{aligned}
\mathcal{P}_q = {} & 48\,352\,414\,907\,668\,482\,133\,492\,087\,869\,890\,625\,q^{42} \\
& + 16\,036\,955\,280\,961\,832\,994\,647\,760\,263\,676\,330\,000\,q^{41} \\
& - 4\,196\,370\,981\,915\,969\,779\,707\,754\,186\,681\,730\,463\,500\,q^{40} \\
& + 1\,218\,782\,850\,654\,754\,001\,200\,323\,954\,657\,938\,014\,382\,400\,q^{39} \\
& - 437\,873\,005\,097\,717\,687\,870\,911\,191\,491\,451\,488\,819\,649\,360\,q^{38} \\
& + 56\,501\,846\,584\,674\,995\,914\,932\,486\,954\,312\,818\,489\,680\,347\,456\,q^{37} \\
& - 1\,286\,142\,494\,794\,109\,681\,758\,225\,357\,376\,327\,215\,945\,929\,878\,016\,q^{36} \\
& - 232\,195\,879\,116\,897\,840\,317\,333\,160\,935\,021\,907\,107\,726\,677\,954\,560\,q^{35} \\
& + 16\,689\,590\,063\,652\,445\,263\,045\,644\,887\,680\,873\,092\,641\,246\,053\,482\,496\,q^{34} \\
& - 348\,554\,024\,754\,220\,147\,603\,324\,828\,210\,793\,549\,927\,603\,425\,605\,058\,560\,q^{33} \\
& - 10\,426\,421\,549\,163\,655\,287\,690\,594\,469\,235\,564\,121\,629\,809\,528\,799\,232\,q^{32} \\
& + 108\,358\,665\,057\,298\,527\,469\,591\,466\,380\,442\,875\,347\,505\,122\,460\,582\,281\,216\,q^{31} \\
& - 1\,768\,689\,203\,525\,174\,631\,731\,070\,128\,293\,659\,678\,465\,684\,698\,618\,344\,243\,200\,q^{30} \\
& + 10\,237\,206\,877\,495\,652\,560\,048\,358\,750\,321\,645\,762\,720\,250\,901\,900\,579\,307\,520\,q^{29} \\
& + 34\,337\,671\,886\,765\,680\,904\,575\,725\,552\,912\,756\,805\,999\,588\,575\,063\,098\,523\,648\,q^{28} \\
& - 893\,036\,607\,748\,838\,369\,727\,901\,354\,793\,161\,228\,395\,689\,789\,685\,496\,497\,045\,504\,q^{27} \\
& + 5\,038\,900\,283\,581\,120\,129\,264\,661\,011\,505\,008\,707\,985\,347\,950\,328\,126\,545\,854\,464\,q^{26} \\
& - 911\,034\,744\,734\,878\,358\,318\,511\,690\,177\,479\,842\,714\,939\,923\,587\,320\,139\,218\,944\,q^{25} \\
& - 136\,638\,994\,950\,831\,706\,692\,876\,722\,748\,596\,049\,298\,348\,362\,299\,318\,950\,265\,094\,144\,q^{24} \\
& + 710\,569\,149\,923\,877\,212\,955\,523\,296\,838\,553\,554\,792\,292\,866\,082\,221\,313\,786\,642\,432\,q^{23} \\
& - 660\,303\,205\,741\,997\,028\,047\,676\,620\,504\,220\,097\,296\,066\,045\,832\,842\,481\,877\,647\,360\,q^{22} \\
& - 8\,445\,211\,998\,036\,033\,041\,924\,146\,120\,899\,793\,736\,566\,787\,600\,428\,875\,407\,043\,854\,336\,q^{21} \\
& + 41\,535\,767\,363\,668\,382\,525\,454\,791\,564\,564\,035\,037\,421\,495\,567\,422\,921\,756\,318\,367\,744\,q^{20} \\
& - 57\,365\,490\,686\,548\,622\,878\,701\,999\,126\,433\,506\,134\,934\,603\,220\,209\,902\,685\,744\,267\,264\,q^{19} \\
& - 188\,825\,222\,514\,258\,821\,335\,193\,309\,495\,064\,592\,166\,989\,341\,589\,042\,262\,945\,921\,236\,992\,q^{18} \\
& + 1\,053\,024\,819\,814\,378\,560\,596\,586\,346\,489\,926\,265\,012\,463\,037\,688\,479\,772\,704\,742\,834\,176\,q^{17} \\
& - 2\,010\,034\,504\,166\,416\,504\,882\,544\,168\,669\,436\,696\,738\,565\,267\,242\,045\,601\,202\,923\,634\,688\,q^{16} \\
& + 241\,622\,642\,955\,207\,095\,961\,297\,706\,882\,489\,064\,894\,767\,543\,964\,612\,636\,166\,202\,589\,184\,q^{15} \\
& + 8\,254\,376\,386\,314\,002\,834\,758\,269\,460\,977\,391\,493\,207\,129\,140\,884\,590\,842\,344\,030\,339\,072\,q^{14} \\
& - 23\,449\,397\,734\,317\,099\,404\,201\,315\,451\,352\,928\,571\,148\,179\,479\,642\,064\,259\,446\,177\,005\,568\,q^{13} \\
& + 37\,752\,878\,249\,181\,220\,083\,251\,025\,160\,951\,030\,076\,509\,596\,020\,273\,002\,567\,883\,600\,953\,344\,q^{12} \\
& - 41\,605\,791\,028\,183\,920\,415\,872\,722\,543\,147\,645\,363\,702\,089\,438\,908\,114\,896\,299\,984\,158\,720\,q^{11} \\
& + 33\,232\,326\,029\,442\,728\,860\,400\,774\,754\,408\,686\,639\,967\,501\,082\,500\,845\,161\,180\,211\,183\,616\,q^{10} \\
& - 19\,695\,752\,677\,516\,758\,371\,623\,100\,757\,223\,744\,077\,407\,681\,308\,451\,195\,377\,189\,902\,090\,240\,q^{9} \\
& + 8\,746\,656\,354\,359\,721\,388\,188\,312\,350\,885\,435\,425\,100\,757\,921\,559\,178\,909\,429\,394\,309\,120\,q^{8} \\
& - 2\,915\,400\,681\,645\,327\,226\,311\,751\,439\,460\,615\,093\,754\,412\,214\,544\,582\,503\,266\,726\,707\,200\,q^{7} \\
& + 725\,633\,915\,347\,957\,804\,608\,976\,221\,187\,775\,000\,761\,066\,750\,692\,966\,098\,974\,801\,920\,000\,q^{6} \\
& - 133\,195\,100\,460\,135\,945\,594\,213\,703\,408\,151\,122\,661\,529\,766\,928\,343\,424\,605\,093\,888\,000\,q^{5} \\
& + 17\,640\,559\,370\,445\,907\,135\,796\,242\,478\,965\,018\,945\,471\,596\,628\,587\,957\,851\,258\,880\,000\,q^{4} \\
& - 1\,626\,421\,616\,965\,037\,836\,715\,783\,050\,899\,480\,429\,100\,567\,881\,451\,399\,702\,118\,400\,000\,q^{3}
\end{aligned}
$$

$$+\,98\,286\,876\,527\,000\,183\,365\,054\,527\,046\,907\,186\,461\,325\,010\,375\,662\,370\,816\,000\,000\,q^2$$
$$-\,3\,475\,143\,775\,680\,439\,103\,880\,620\,233\,736\,199\,312\,494\,519\,626\,361\,733\,120\,000\,000\,q$$
$$+\,54\,165\,092\,504\,344\,744\,846\,141\,769\,155\,192\,828\,275\,753\,259\,892\,736\,000\,000\,000\,. \tag{30}$$

The fixed point value of $g_{D\phi^4}$ (as well as the fixed point values of all gamma functions) can then be expressed in terms of a polynomial in $g$ of order 41 evaluated at the corresponding fixed point value for $g$, whose coefficients are too large to display here – for $g_{D\phi^4}$ these coefficients are ratios of integers with each having about 700 digits. The expressions for the fixed point value of $g_{D\phi^4}$ and the gamma functions can be found in the Mathematica notebook [145].

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
