# Peer review of "Safe essential scalar-tensor theories"

_SciPost Physics_

## Round 2 · Referee Report · Anonymous (Referee 1) · 2024-5-1

Weaknesses

1-This paper lacks some crucial details regarding calculations and definitions.

Report

Dear editor and author,

I have reviewed the paper titled “Safe essential scalar-tensor theories”. In this paper, the author studies the asymptotic safety scenario for scalar-tensor theories with shift symmetry by using the functional renormalization group in the essential scheme. This scheme is a recent development and allows us to remove the redundant (or inessential) operators by the equations of motion such that only “essential” operators are taken into account in the effective action. The fixed point structure and the renormalization group flow are investigated.

While the paper shows interesting results, some details are omitted too much and the paper is dedicated only to a few experts in the asymptotic safety community. This becomes an obstacle to read the paper for non-experts. I suggest several modifications which are listed below.

(i) I cannot see how the essential operators are selected in the effective action (4). Although Footnote 2 describes a brief description, it is still difficult to fully understand it. I think it is better to summarize the process to determine the essential part and inessential part, e.g., in an appendix.

(ii) Above Eq (16), “minimal” is emphasized. The author should mention what the meaning of “minimal” is.

(iii) Mention the definition of critical exponents in order to avoid confusion because some literatures use the convention that relevant operators have negative critical exponent.

(iv) In the paper, the explicit form of the beta functions is totally omitted. I guess those take very lengthly forms, while some explanations for the flow equations are helpful to understand the result. In particular, the schematic structure of the beta functions for obtaining the fixed point values A, B, C (and Gaussian fixed point D) should be presented. The diagrammatical representation for the beta functions would be also helpful.

(v) Clarification is needed regarding theta_1 and theta_2, which correspond to the critical exponents of g and g_{D\phi^4}, respectively even if it is apparent from the dimensional counting. Besides, the operator mixing leads to the different definition of the critical exponents at non-trivial fixed points from those at the Gaussian fixed point. Therefore, the author should use different notations for theta_1 and theta_2.
Eq. (17), theta_1 represents the critical exponent of D\phi^4 and theta_2 represents that of the Newton coupling, while in Eq. (18), theta_1 corresponds to the critical exponent of the Newton coupling and theta_2 to that of D\phi^4. This inconsistency could potentially confuse readers.

(vi) Tables 1 and 2 contain extensive data. I believe that displaying all fixed points may not be necessary for the ensuing discussion. I propose relocating these tables to Appendix A and showing only fixed points A and B in the main text.

(vii) In Eqs.(18) and (19), it is difficult to estimate the magnitude of the fixed point values, the critical exponents and the gamma functions at first sight. Therefore, I recommend to write their decimal values as well as the exact values.

(viii) In the end of Section 3.1, the author mentions that only three of the interacting fixed points are reasonable as globally well-defined fundamental QFTs. How to arrive at this conclusion? What is the criterion for judging whether a fixed point is globally well-defined fundamental QFT or not?

(ix) The author refers to the positivity bound discussed below Eq.(22). However, in Euclidean spacetimes, the trace mode has a “wrong”-sign kinetic term as evidenced, for instance, by the regulator applied for the trace mode in Eq. (9). Notably, while the other fields possess a positive sign in their regulators, the trace mode manifests a negative sign.) Consequently, the assurance of action positivity becomes incomplete. Besides, the scalar field \phi and the trace mode h are the same degrees of freedom, so they mix. In light of this interplay, how can one give significance to the positivity bound for the scalar field in this setup?

(x) The ”weak gravity bound” in this paper seems to be different from that of the so-called “weak gravity conjecture” [N. Arkani-Hamed, L. Motl, A. Nicolis and C. Vafa, JHEP 06 (2007) 060 [hep-th/0601001]]. The author should clearly explain what the definition of the weak gravity bound of this paper is.

(xi) In Section 3.3, the author introduces \alpha and \beta as gauge fixing parameters without their definition.

(xii) The author mentions a relation between shift-symmetric scalar field and string theory. What are implications of the obtained result in the paper to the string theory then?

Minor point:
What is a dot on \Phi in the left-hand side of Eq.(2)?

Recommendation

Ask for major revision

---

## Round 2 · Referee Report · Anonymous (Referee 2) · 2024-5-23

Report

The paper studies shift symmetric scalar field with quantum gravity in the asymptotic safety scenario. The novelty is that the system is studied in the essential scheme, which allows for eliminating inessential operators via field redefinitions. This simplifies the computation by reducing the number of operators at each order of the truncation. The paper is clearly written (up to minor clarifications) and presents an important step in the understanding of scalar-gravity systems. I recommend its publication after some minor clarifications are implemented.

I will not repeat points that have already been mentioned by Referee 1. I have the following additional comments:

(i) I would appreciate a discussion of the physics associated with the different fixed points. Which of the fixed points can be trusted to exist beyond this truncation? For example, fixed point C provides a UV completion of scalar theories without gravity. This contrasts the usual expectation of scalar theories having a triviality problem. The author has a follow-up paper (Ref. [148]), which discusses some of these aspects, but I would nonetheless appreciate a discussion to make the paper self-contained.

(ii) I am happy that the author provides a supplementing Mathematica notebook which includes the beta and gamma functions as well as expressions for the traces. To maximise the impact of the notebook, I suggest including some minimal clarifying comments. Especially the section 'RHS of flow equation for general regulator' is hard to understand without any explaining comments.

(iii) I don't see the added value in displaying the polynomial in Appendix A. Instead displaying the beta functions, which seem to be of similar size according to the Mathematica notebook, would be much more useful.

(iv) The ghost action in equation (8) has a rather unusual normalisation with $\frac1{\sqrt{G_N}}$, which gives the ghost a mass dimension of 1/2. Is there any reason for this choice, which might confuse some readers?

Recommendation

Ask for minor revision

---

## Editorial Decision

awaiting_resubmission